# Meta-Governance Framework to Guide the Establishment of Mass Collaborative Learning Communities

**Majid Zamiri \*, Luis M. Camarinha-Matos** 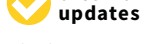 **and João Sarraipa**

School of Science and Technology and Center of Technology and Systems (CTS-Uninova), NOVA University of Lisbon, 2829-516 Monte de Caparica, Portugal; cam@uninova.pt (L.M.C.-M.); jfss@uninova.pt (J.S.)
\* Correspondence: ma.zamiri@campus.fct.unl.pt

**Abstract:** The application of mass collaboration in different areas of study and work has been increasing over the last few decades. For example, in the education context, this emerging paradigm has opened new opportunities for participatory learning, namely, "mass collaborative learning (MCL)". The development of such an innovative and complementary method of learning, which can lead to the creation of knowledge-based communities, has helped to reap the benefits of diversity and inclusion in the creation and development of knowledge. In other words, MCL allows for enhanced connectivity among the people involved, providing them with the opportunity to practice learning collectively. Despite recent advances, this area still faces many challenges, such as a lack of common agreement about the main concepts, components, applicable structures, relationships among the participants, as well as applicable assessment systems. From this perspective, this study proposes a meta-governance framework that benefits from various other related ideas, models, and methods that together can better support the implementation, execution, and development of mass collaborative learning communities. The proposed framework was applied to two case-study projects in which vocational education and training respond to the needs of collaborative education–enterprise approaches. It was also further used in an illustration of the MCL community called the "community of cooks". Results from these application cases are discussed.

**Keywords:** mass collaborative learning; meta-governance framework; mass collaboration assessment



## 1. Introduction

Learning has taken place since before the earliest civilizations were formed. Reviewing the practices used for learning throughout history up to the present day shows that learning methods have evolved gradually but surely. Even though learning methods have come a long way, looking at the future of learning requires making changes to such methods to keep them up to date with the context and growing demands of the 21st century [1]. In this regard, mass collaborative learning (MCL) as an emerging method has proved to be beneficial and supportive for autonomous learning. MCL unlocks new opportunities for an unlimited number of scattered learners to build up a self-directed community and achieve an entirely new level of flexibility, efficiency, and customization. The design of this flexible learning space goes beyond the physical context to include virtual or mixed reality. That is, MCL, as a form of open-source community and as a flexible learning environment, can potentially take place on diverse platforms such as mixed reality, virtual reality [2], social virtual reality environments (immersive virtual worlds or multi-user virtual environments) [3], multi-user 3D interactive environments, and 3D multi-user virtual worlds (e.g., social virtual worlds, open-source virtual worlds, and collaborative virtual learning worlds) [4]. Thus, the concept is leaping into the digital age, to be supported and enriched by ICT technologies. In the informal process of learning that is implicit in MCL, cross-functional collaboration, peer-to-peer learning, and crowdsourcing are the predominant strategies. As this new trend of learning continues to grow, it will require preparation for changing the nature of learning practices [5,6].

MCL opens the possibility for the general public to return to learning habits, anytime and anywhere. That is, MCL fosters lifelong learning after formal education is complete and throughout life. It foresees a learning-friendly environment where learners are encouraged to join a community and actively contribute to a vast range of defined activities and topics. It not only enables the participants to engage in collaborative knowledge acquisition, creation, sharing, retention, and development but also allows them to be curators of knowledge. Hence, participants become independent learners who are leaders of their own education [7]. There are many potential benefits from the application of this complementary method of learning. For instance, it creates a specific network of enthusiastic participants—from pupils and nonscientists to professionals and experts—who may come from different backgrounds; when it comes to solving complex problems (that are mainly beyond one's own ability), strategies such as critical thinking, group discussion, voting, collective intelligence, and outsourcing will help all the learners involved [8,9].

The success that mass collaboration has achieved over the last few years in the learning domain is, in fact, noticeable. The literature shows that mass collaboration has morphed and expanded the physical, virtual, and intellectual boundaries of learning environments [10]. As such, mass collaboration has had many successful applications (at different levels and to different degrees) in diverse learning contexts, for example:

- Massive open online courses (MOOCs), which are free online courses where the learning contents are delivered to any person who wants to take a course. In this model of learning, one that is designed for large numbers of geographically dispersed students, they can practice learning individually (in personal tasks; thus, this is not really mass collaboration) and through community interactions (via interactive courses, featuring a form of mass collaboration).
- World of Warcraft (WoW), which is a massively multiplayer online role-playing game that facilitates learning through gamification. WoW creates a community of players where participants can play with others in temporary groups. In this collaborative space, learning occurs when the learner needs and wants it. Therefore, in the context of problem-solving, there are opportunities to receive the answer (from other players or more experienced peers) to a question or obtain advice quickly.
- Scratch, which is a free programming language and online community where scratchers/learners can create their own interactive stories, games, and animations. Scratch promotes problem-solving skills, self-expression, collaboration, and creative teaching and learning. There is a discussion page with multiple forums mainly used for chatting, helping (with coding), creating, sharing, and learning together [11].
- The SAP community network (SCN), which is a community of software users, developers, consultants, mentors, and students who use the network to get help, share ideas, learn, innovate, and collaborate [12].
- The community of inquiry (Col) framework is a community of learners and instructors who share a virtual space, technology-reliant environment, rule-based interaction, and course-dependent learning objectives that resulted from the interaction of the perceptions of social, cognitive, and teaching presences [13].
- The ePals global community, which is an example of social online learning that provides the necessary tools (platform) and meeting places to build a worldwide community of learners, global citizens who can share ideas, practice communication, and offer help and guidance [14].

Despite such promising achievements and the great opportunities that MCL has already opened up for learners, communities, and societies, accessing this body of knowledge still faces several challenges. The novelty of the MCL concept, the complexity of its underlying processes, and the interdisciplinary nature of the method have all intensified the identified issues. In particular, there is no widely agreed definition of "MCL", nor have all its aspects, characteristics, and components been explicitly defined as yet. Furthermore, this approach of collective learning is viewed and presented from different perspectives by the researchers of each discipline [6,8,15]. There is no doubt that, without a clear understanding

of this subject, it is extremely difficult to design an appropriate governance structure by which an MCL community can be built, controlled, operated, and developed.

Given the above, it is necessary to raise the importance of a comprehensive governance structure for MCL that provides a clear understanding and oversight about how a learning community can be directed and managed as well as how its objectives could be set and achieved. Therefore, the main contribution of this article is proposing a meta-governance framework for an MCL community (MGF-MCL), aimed at driving and supporting its implementation, operation, and management. The MGF-MCL consolidates the integration of ideas from different governance styles (some organizational, behavioral, and governance structures, as well as a set of assessment methods) to achieve an effective outcome.

The remainder of this paper is organized as follows: Section 2 presents background information in brief, Section 3 explains the adopted research method, Section 4 presents the proposed meta-governance framework with detailed information, Section 5 discusses the importance of the proposed framework in terms of an MCL community, and briefly looks into possible future work. Finally, conclusions are given in Section 6.

## 2. Background Information

The idea of practicing collaborative learning on a large scale was first introduced by Illich in 1971 [16]. Surprisingly, 25 years before the Internet was invented, he conceptualized just such an educational approach, where interested people can access available resources for learning at any time in their lives. Illich envisioned an equipped learning environment in which educators can freely exchange their knowledge and skills with peers. From that time onward, numerous advances in science and technology have driven the evolution of MCL and enabled it to gradually grasp more and more helpful features and capabilities needed for effective learning. For instance, the application of computer-supported collaborative learning tools has enabled the MCL communities to create innovative socio-technical environments where learners can collectively and/or independently work on specific modules and topics, facilitating innovative and decentralized learning [12,17]. The latest developments (e.g., various sophisticated communication channels [5], the ability to solve complex problems [7], and the increased quality of learning materials [8]) in current examples of MCL (e.g., some cases of MOOCs, Scratch) have influenced the education and learning ecosystem to a great extent. As such, the shreds of evidence show that the MCL system has been fairly successful in participatory knowledge acquisition, creation, preservation, sharing, assessment, and development [10,11].

Despite the advantages of traditional face-to-face and in-person learning, it faces some limitations or difficulties, for example, travel time and costs, fees, scheduling, low flexibility, etc. In contrast, in the MCL model, the collaboration and interactions predominantly take place over the Internet; this not only hinders the development of such limitations and difficulties but also MCL can be used as a flexible learning solution when face-to-face learning is impossible or poses a risk. This is particularly relevant in current conditions, where the COVID-19 pandemic is causing alterations in educational systems and a (temporary and compulsory) shifting away from the classroom to alternative modes of learning (e.g., virtual) [18]; MCL can respond to the current pandemic and create a long-term positive impact by providing support for every student/citizen around the world. As an illustration, MCL can provide an innovative and suitable environment with specific features (e.g., safe, customizable, affordable, conventional, flexible, and accessible), equipped with modern technologies (e.g., video conferencing, blogs, discussion forums, or platforms) through which a large number of interested people (e.g., teachers, students, and even general public) at regional, national, or international level can engage in collaborative learning practices (e.g., sharing learning materials, discussion) throughout the crisis and even after.

MCL communities are typically self-governed, non-hierarchical, heterogeneous, and dynamic. In such communities, large groups of learners who are geographically, temporally, and conceptually dispersed participate in and contribute to diverse tasks on a voluntary basis [6]. Generally, these comprise "diverse sets of autonomous learners (with various

levels of expertise and performance) aiming to reap the power of collaboration and the advantages of diverse minds toward learning . . . favorite subjects" [8]. Even though the principles of an MCL governance system point to democratic learning communities and fostering social inclusion, related contributions in the literature are fragmented and there are still unclear issues regarding its organizational mechanisms, including organizational and behavioral structures [7,10]. An organizational structure outlines (a) what kind of roles can be performed, (b) how certain activities should be directed, (c) how arrangements and relationships can be developed, and (d) how information might flow between levels within an organization to reach common goals. A behavioral structure delineates the personal aspects (personality, attitude, and motivation), social aspects (collaboration, communication, and contribution), and behavioral patterns (also concerned with the assignment of responsibilities) of involved entities in the organization. Furthermore, the behavioral structure reflects the principles, policies, governance rules, and decision-making policies that either steer or confine the behavior of the organization and its members [19]. Indubitably, all organizations like MCL communities require an appropriate structural and behavioral framework and adaptations to help them survive and thrive.

Even though MCL communities have been increasingly used for a range of public benefits, it is still the case that little is known about what are the major factors and variables that constitute governance, how they interact, and why they are relevant [20]. In the context of a complex system like MCL (with an increasingly growing environment), past research in the literature has viewed its organizational, behavioral, and governance structure from different perspectives. Additionally, the principles, processes, procedures, elements, and rules associated with such structures have been defined in diverse ways. For example, the authors of [21] declare that prior conditions (e.g., legal, political, environmental, socioeconomic, technological, and regulatory elements) influence the formation of a community and the development of its governance structure. Another study [20] highlights the role of several critical processes (e.g., leadership, administration, construction of legitimacy, and confidence-building) that influence the day-to-day management and operation of collaborative communities. In [22] the authors present a categorization of the factors affecting the governance structure, according to two main groups: (a) internal factors, dealing with a complex set of problems that are internal to an organization (e.g., different types of shareholders, conflicts of interest between shareholders and managers, and block-holder opportunism), and (b) external factors, that manage the relationships with external parties (e.g., community members, alliance partners, and overseas subsidiaries). Other studies [15,19] emphasize the roles of internal aspects (e.g., structural, componential, functional, and behavioral dimensions) and external interactions (e.g., market, support, societal, and constituency dimensions). An analysis of the literature reveals different aspects of the influential factors and contributing elements regarding the governance structure of MCL. To streamline the identification of the associated elements, it is better to look at the governance structure holistically as a system of interdependent but harmonious elements.

In addition to the governance aspects and related components, the MCL community also needs to keep an eye on the assessment aspects to measure, compare, and analyze the coherence between results and objectives (both general and specific). Assessment of a learning community or collaborative learning is a planned systematic review that focuses on understanding, identifying, collecting, and analyzing the data related to the dynamics at work, such as operational activities and processes. The assessment process helps the learning communities to find out the specific needs, strengths, weaknesses and learning gaps that need consideration [23]. Furthermore, assessment can provide a good indicator of the degree of community development, progress, and effectiveness from different points of view (e.g., organizational, social, and behavioral) [24,25]. In the context of assessment, the literature has generally followed two main approaches. The first approach focuses on identifying the common approaches for assessing community and collaborative learning. For example, the potential procedures (e.g., individual assessment, group assessment, or group assessment combined with intra-group peer assessment [26]), the effective methods

(e.g., summative and formative [26,27]), and the recognized techniques (e.g., machine learning techniques [28]) that can be used to assess collaborative learning at different levels of community and at any given time. The second approach highlights the importance of community outcomes and collaboration "products" including but not limited to individual, group, and community performances [28,29], specific abilities such as knowledge creation and sharing [17,27], and high-quality contents [8], and learning progress [30]. The MGF-MCL, although inspired by those previous developments, tries to give further insights into an integrated perspective of the most complex concerning issues and also some internal and external aspects of corporate governance in the MCL community.

## 3. Research Method

### 3.1. Overall Research Method

This work results from Ph.D. thesis research on mass collaboration and learning. For this study, the design science research process (DSRP) approach [31] is adapted to (a) concretize our research design, and (b) develop and validate the proposed framework (MGF-MCL).

The DSRP paradigm has its roots in engineering and is basically used for the purpose of problem-solving. DSRP aims to develop organizational and individual capabilities by the creation of innovative artifacts and the generation of design knowledge. The method comprises six main steps [32]:

- *Problem identification and motivation*: identifying the research problems and justifying the value of the solutions. As mentioned in the Introduction, this work faces some of the main challenges, such as topic complexity, insufficient structured information (in the literature) about different aspects of MCL. Thus, this step contributes to (a) identifying the main components, features, and characteristics of MCL communities, (b) finding the structural and behavioral aspects of MCL communities, and (c) addressing potential indicators for assessing learning and performance.
- *Definition of the objective of the solution*: defining how the identified problems should be solved. As pointed out in the Introduction, this study strives to highlight the significance of supporting the processes of implementation, operation, and management of MCL communities. To understand, summarize, and synthesize the existing research, debates, and ideas around this body of knowledge, we conducted a deep literature review. This review helped us to provide a foundation of (general and specialized) knowledge on the topic.
- *Design and development:* designing an artifact (e.g., model, framework) that can solve the problems detailed above. In responding to the identified problems (addressed in the prior step) and research questions (listed in Section 3.2), we proposed the MGF-MCL framework (presented in Section 4). MGF-MCL is the main contribution of this work. The framework emanates from background knowledge and experiences, in combination with some relevant solutions reported in the literature (and mentioned in part in Section 2).
- *Demonstration:* finding a suitable context by which to demonstrate the usefulness of the artifact. To measure the success of MGF-MCL in supporting and directing the MCL communities, the framework was applied to three different case studies (presented in Section 5).
- *Evaluation:* evaluating and observing how well the artifact works. In order to evaluate how successful the MGF-MCL is in supporting and steering the target case studies, we used different assessment methods and processes. For example, through a three-phase evaluation process, we have assessed the acceptability, capability, and effectiveness of the framework in the case studies (see Figure 1).
- *Communication:* reporting the effectiveness of the proposed solution (framework). The inputs and outputs of this work are shared with others through publication.

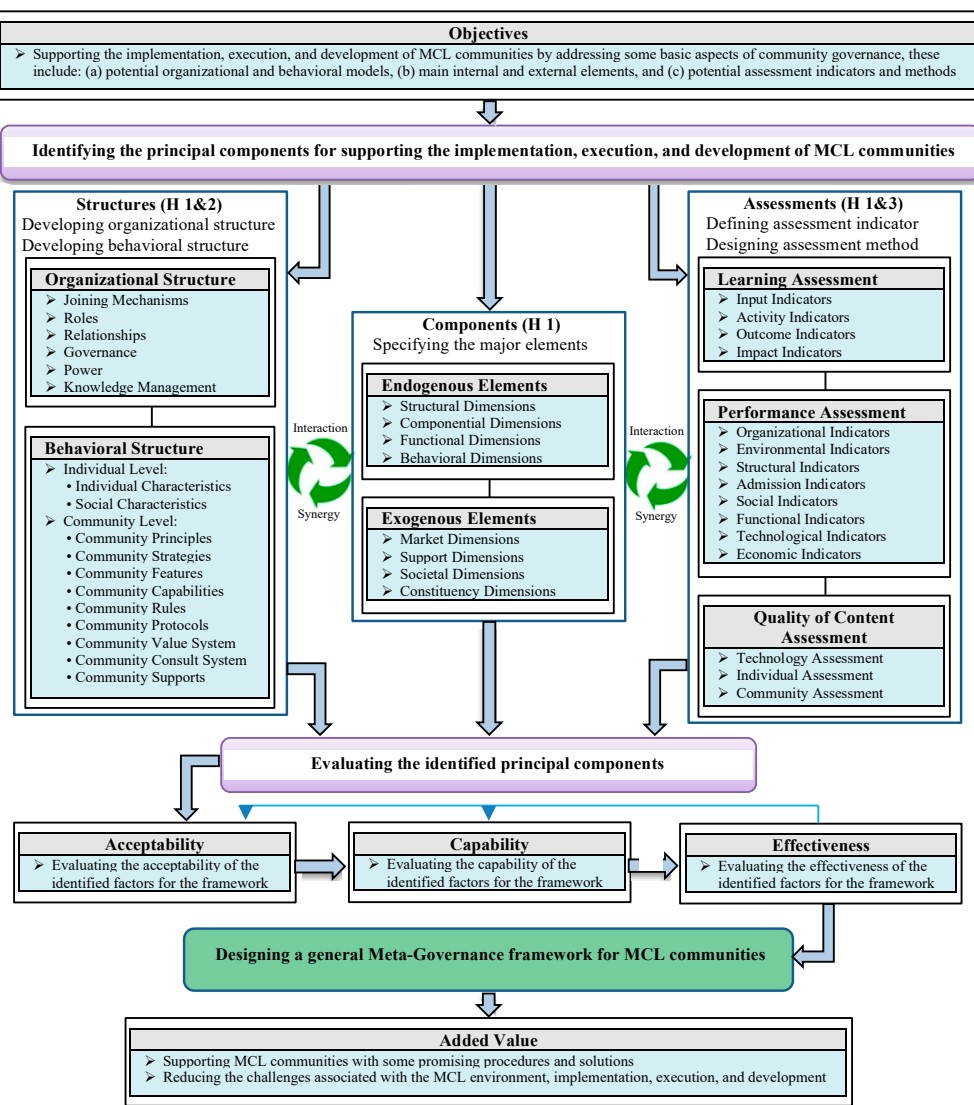

**Figure 1.** Meta-governance framework for MCL communities.

*3.2. Research Questions and Hypotheses*

To appropriately guide the research, three research questions were initially formulated:

**RQ1.** *What could be an effective way of supporting community learning through mass collaboration?*

**RQ2.** *What kind of organizational structure within a community should be established to help develop learning through mass collaboration?*

**RQ3.** *What kind of assessment mechanism can help minimize the problems related to the reliability of created and shared knowledge or information through mass collaboration within a community?*

The following hypotheses are then set to respond to the above-mentioned research questions:

**Hypotheses 1 (H1).** *Community learning could be effectively supported through mass collaboration if three streams of work are appropriately rooted in the foundation of a community; namely, through (i) identifying the positive and negative factors in existing and emerging successful examples of mass collaboration; (ii) adopting contributions from collaborative networks in terms of structural and behavioral models; and (iii) establishing adequate learning assessment indicators and metrics.*

**Hypotheses 2 (H2).** *Community learning through mass collaboration could be supported if existing models of organizational structures for long-term strategic networks are extended to allow*

*more fluid borders, and new roles, incentives, and internal subgroups are defined to focus on learning and knowledge generation.*

**Hypotheses 3 (H3).** *The problems related to the reliability of created and shared knowledge or information through mass collaboration could be minimized not only if the community benefits from the combination and application of a set of appraisal rules, criteria, and methods but also if the content materials are critically assessed through a collective effort.*

### 3.3. Search Strategy

As an initial step, a systematic literature review was performed to provide a comprehensive overview of the domain, basic concepts, affecting factors, and required structures for MCL communities, as well as to identify the contradictions and gaps in the related literature. A fundamental eligibility criterion for selecting studies in the review was that they could relate to the formulated research questions. The search was performed on databases such as SCOPUS, IEEE Xplore, Web of Science, and Google scholar, resulting in the selection of 253 articles within the period of 2010–2021. After reviewing the keywords, abstracts, and conclusions for relevance, 157 full articles were selected for reading. Upon narrowing down the selection, a total number of 101 articles were eventually included in the final analysis [5].

### 3.4. Inclusion and Exclusion Criteria

For this work, it was first necessary to determine the inclusion and exclusion criteria for the study. Thus, the original journal articles, book chapters, surveys, conference materials, and technical reports that were published in the English language were included. The search strategy also includes the selected articles' reference lists. This means that if we found authors who made good contributions, we checked their other related publications. Other types of documents, such as conference abstracts, editorials, position statements, expert opinions, comments, and letters were excluded.

### 3.5. Keyword Selection

To find the relevant papers by searching the database (for title, abstract, and body text), a range of keywords (both alone and in combination) were used. For each element of the framework (addressed in Figure 1), the search terms are as follows:

- *Organizational structure*: "organizational structure of collaborative communities/ networks/learning communities/learning initiatives";
- *Behavioral structure*: "behavioral structure of collaborative communities/ networks/learning communities/learning initiatives";
- *Endogenous elements*: "endogenous elements of collaborative communities/ networks/learning communities/learning initiatives";
- *Exogenous elements*: "exogenous elements of collaborative communities/ networks/learning communities/learning initiatives";
- *Learning assessment*: "assessing learning process in collaborative communities/ networks/learning communities/learning initiatives";
- *Performance assessment*: "assessing the performance of collaborative communities/ networks/learning communities/learning initiatives";
- *Quality of content assessment*: "assessing the quality of contents in collaborative communities/networks/learning communities/learning initiatives".

### 3.6. Categorization

The selected papers were grouped and classified into categorical themes (e.g., base concepts, related organizational and behavioral structures, main internal and external components, potential assessment methods, and assessment indicators and metrics). The most important data (e.g., major topics, objectives, findings, open issues, remarks, and references) were then extracted, tabulated, and documented to be used in the work.

*3.7. Outcome Measures*

The papers were then analyzed according to the above-mentioned guiding research questions. Next, the structures of the chosen works were analyzed step-by-step. For example, the research methods used in the studies were reviewed to identify whether or not they appeared to be suitable. The supporting pieces of evidence and facts were carefully checked. Additionally, the sources that were used by the authors were analyzed to get a better idea of how they had formed their thoughts. The collected data were qualitatively and quantitatively evaluated. A summary of these findings can be found in [5,7,8,10,15]. The current article not only integrates results from this study but also combines and summarizes the findings of the thesis work and presents them in a single framework (meta-governance) to be used as a guide for establishing and developing mass collaborative learning communities.

The results of the other steps of the research method are presented in the following sections.

## 4. Proposed Meta-Governance Framework

This section first introduces an overall picture of the meta-governance framework, followed by more detailed information about the various parts of the framework. In this paper, the meta-governance framework is considered as a comprehensive, organized, and specified structure for use by MCL initiatives. The proposed meta-governance framework is a generic and conceptual structure intended to highlight the main concepts, factors, and connections associated with MCL communities. The meta-governance framework integrates ideas from different governance styles (e.g., participatory or democratic governance, corporate governance, and network governance) and then consolidates the key components, attempting to introduce a structure (with mixed dimensions) that can serve as a guide for supporting MCL communities. For example, there should be a set of processes, functions, and activities (e.g., assessments, social participation, interactions) in each MCL community that should be properly carried out to meet and accomplish the goals of the community. For this reason, the meta-governance framework has been developed, aiming to portrait, guide, and streamline such particular tasks.

It is noteworthy that the meta-governance framework and its containing elements and features are defined to be used on a mass scale; however, it is possible to scale them up or down for learning communities of any size. The fact is that since meta-governance is a dynamic framework, it should then be adapted according to the objectives, requirements, and conditions of the concrete case. Considering the above hypothesizes, the framework includes three main interconnected parts:

A. *Structure Part*: refers to the type of structures that can be used for building, arranging, and organizing the MCL community. It comprises two types of structure:

- Organizational structure: outlining how certain activities are delegated toward achieving the goals of the MCL community.
- Behavioral structure: identifying the behavioral patterns and culture of the MCL community.

This part covers the first and second research hypotheses.

B. *Component Part*: addresses the main internal and external components of the MCL community:

- Internal components: focusing on the identification of the main set of elements/properties that can together describe the internal environment of the MCL community.
- External components: focusing on the interaction of the MCL community with its external environment.

This part covers the first research hypothesis.

C.   *Assessment Part:* emphasizes the importance of community evaluation and, thus, provides a picture of community changes, both positive and negative. It consists of three assessment principles:

- Learning assessment indicators: addressing the main indicators that reflect the individual's learning.
- Performance assessment indicators: highlighting the main indicators to assess the community performance.
- Assessing the quality of content: identifying potential methods to assess the quality of the content shared among individuals within the community.

This part covers the first and third hypotheses.

It should be noted that these three parts of the framework are not only not separable in reality but also have different levels of support, interaction, and synergy that strengthen the framework and increase its potential value.

As shown in Figure 1, the main objectives of the meta-governance framework (derived from the goals of this research) are first addressed. Fulfillment of the objectives can help in achieving added value for the MCL community. The framework was developed in two main parts:

- *Part 1*—identifying the principal components necessary for supporting the implementation, operation, and development of MCL communities. This part focuses on three main tasks:

  - Identifying the potential organizational and behavioral models, aiming at clarifying the community scaffold, scope, authority, and human resources.
  - Identifying the main internal and external elements to raise awareness of the related concepts, environments, entities, relationships, and interactions.
  - Identifying the potential assessment indicators and methods for gathering relevant information about positive and negative changes, such as each individual's learning progress, community performance, and the distribution of low-quality materials within the community.

- *Part 2*—evaluating the identified components against the requirements, conditions, and objectives of a target MCL community. This means that when the meta-governance framework is adjusted for application to a specific MCL community, it then needs to be evaluated to ensure that it can meet the expectations and objectives of that community. The evaluation process proceeds in three main phases: namely, evaluation of the acceptability, capability, and effectiveness of the framework. This process can help to appropriately measure, develop, and validate the framework for the target community. The process of evaluation can be taken over by internal members of the community or be outsourced. Depending on community strategies, various methods can be used for evaluation, such as the Delphi method.

### 4.1. Structural and Behavioral Models

Every organization (community) requires an appropriate structure (even in the case of self-organized communities, like MCL) to survive, operate, and grow. The structural model should be selected based on the community's needs, priorities, and objectives. As communities nowadays are moving from physical-based to digital-based operation, their structures are shifting accordingly from the hierarchical and centralized model toward a bureaucratic and collaborative model [10]. In the MCL context, the structural model delivers an informative representation of the community's joining mechanisms, participants, roles, relationships among the roles, and governance approaches. The proposed structural model (presented in Table 1) is inspired by the related organizational structures and models for collaborative communities [10,20–22] reported in the literature.

**Table 1.** General structural model for MCL communities.

| General Structural Model for MCL Communities | | | |
|---|---|---|---|
| **Joining Mechanisms**<br><br>➢ **Applicant**:<br>  • Sends application for joining<br>➢ **Community**:<br>  • Accepts the application, or<br>  • Rejects the application, or<br>  • Requests correction<br><br>**Joining Conditions and Rules**<br><br>➢ **Inclusion & Exclusion**:<br>  • Easy inclusion and exclusion<br>  • Open access for all<br>  • Free of charge<br>➢ **Accessibility**:<br>  • Registration is mandatory<br>  • Registration with real information<br>  • Minimized anonymity<br><br>**Level of Engagement**<br><br>➢ **Type of Groups**:<br>  • Group of ordinary members<br>  • Group of experts<br>  • Group of technical members<br>➢ **Type of Members**:<br>  • Visitors<br>  • Active members<br>  • Inactive members<br>➢ **Type of Engagement**:<br>  • Knowledge creation<br>  • Knowledge sharing<br>  • Knowledge development<br>  • Knowledge assessment | **Roles**<br><br>➢ Managerial Roles:<br>  • Identity controllers<br>  • Content controllers<br>  • Administrators<br>  • Moderators<br>➢ Technical Roles:<br>  • System managers<br>  • IT technicians<br>  • Technical operators<br>  • Technical support<br>➢ Participatory Roles:<br>  • Experts<br>  • Ordinary members<br><br>**Relationships Among Roles**<br><br>➢ Relationship Rules:<br>  • Based on mutual trust<br>  • Relies on collaboration<br>  • Can build friendship/s<br>  • Can create group/s<br>  • Can create discussion/s<br>  • Can extend to outside<br>➢ Relationship Types:<br>  • Short term<br>  • Long term<br>  • Formal<br>  • Informal<br>  • Intimate<br>  • Superficial | **Governance**<br><br>➢ Governance Focus:<br>  • People<br>  • Purposes<br>  • Policies<br>  • Processes and Procedures<br>  • Participations<br>  • Performances<br>➢ Governance Rules:<br>  • Developing transparency<br>  • Emphasizing responsiveness<br>  • Encouraging accountability<br>  • Increasing effectiveness<br>  • Reducing risks<br>  • Following fairness<br>  • Consensus Oriented<br>➢ Type of Governance:<br>  • Self-governed<br>  • Collaborative<br>  • Encouraging accountability<br>  • Democratic<br>  • Non-hierarchical<br>➢ Governance Borders:<br>  • Self-governed<br>  • Internal interactions<br>  • External interactions | **Knowledge Management (KM)**<br><br>➢ KM Components:<br>  • People<br>  • Procedures and Methods<br>  • Contents<br>  • Strategies and Tools<br>➢ KM Processes:<br>  • Collecting<br>  • Organizing<br>  • Summarizing<br>  • Analyzing<br>  • Synthesizing<br>  • Making decision<br>➢ KM Approaches:<br>  • Knowledge is continually created, shared, and developed<br>  • Knowledge is continually turned from tacit into explicit form<br>  • All members are responsible for quality assurance<br>  • The quality of knowledge is continually assessed and improved<br>  • Well-developed knowledge will be sorted, stored, and used for learning<br>  • Learning occurs at both individual and community levels |

The behavioral model represents a set of related features that have a direct or indirect impact on the behavior of the MCL community. The proposed behavioral model in this work embraces two main parts:

- *Behavioral features at the individual level*: these reveal those personal attributes (e.g., personal characteristics, personal skills, and background information, as well as individual capacity) that have a significant effect on individual and community learning.
- *Behavioral features at the community level*: these focus on a set of community features and capabilities that play a key role in the performance and success of the community from the behavioral point of view.

The proposed behavioral model is presented in Table 2.

**Table 2.** General behavioral model for MCL communities.

| General Behavioral Model for MCL Communities | |
| --- | --- |
| **Individual Level** | **Community Level** |
| ➢ Personal Characteristics:<br><br>• Personality<br>• Attribute toward collaboration<br>• Desire to collaboration<br>• Cognitive abilities<br>• Commitment<br>• Motivation<br>• Awareness<br>• Competence fitness<br>• Perceived usefulness<br>• Perceived ease of use<br><br>➢ Personal Skills and Information:<br><br>• Having the necessary information<br>• Having necessary skills<br>• Having the necessary experience<br><br>➢ Person's Capacity:<br><br>• Ability to understand information<br>• Ability to use information<br>• Ability to communicate<br>• Ability to make decision<br>• Ability to act under the law | ➢ Community Features:<br><br>• Community strategies<br>• Community rules<br>• Community's behavioral patterns<br>• Community's leadership principles<br>• Community's decision-making principles<br>• Community's brokering principles<br>• Community's value systems<br>• Community's motivation approach<br>• Community's rewarding policies<br>• Community's trustworthiness<br>• Community's preparedness<br>• Community's feedback system<br>• Community's agreements and negotiation approach<br>• Community's partnership strategies<br>• Community's consult system<br>• Community's support system<br>• Community's cultural influences<br>• Community's social influences<br>• Community's conflict resolution approach<br>• Community protocol |

It is noteworthy to mention that modeling complex systems, such as MCL communities, under an interdisciplinary approach requires a clear and profound understanding of the concepts involved (the disciplines of collaborative network and learning) to capture their complexity. The proposed behavioral model (Table 2) is inspired by certain related structures [33] and models [10,34]. Furthermore, the addressed components, non-tangible features, and characteristics listed in both the structural and behavioral models are general and dynamic. In other words, when applying them in a specific scenario of MCL, they need to be accordingly and appropriately modified, based on the objectives, requirements, conditions, and resources of the case.

### 4.2. Internal and External Elements

For the creation and development of MCL communities, in addition to the structural and behavioral models (presented above), a comprehensive set of concepts, entities, and affecting factors should be taken into account to cover both the internal and external aspects of the surrounding environment. However, capturing and structuring a wide range of general and specific related concepts and entities is not an easy task, and it needs a long-term commitment. In order to elucidate its inherited complexity and take the first step in this process, we propose a reference model for MCL communities (MCL-RM) [15] that represents the involved environment features, specifications, and interactions: namely, internal elements (endogenous elements) and external elements (exogenous interactions). The MCL-RM model uses ideas from the related studies published [15,19] and was inspired by the ARCON modeling framework [15,35]. The endogenous elements focus on controllable entities, properties, functions, and features of the MCL community. Endogenous elements comprise four dimensions, namely:

• *Structural dimension*—refers to actors (participants) in the community, and their roles and inter-personal relationships.

- *Componential dimension*—refers to those physical and tangible resources (e.g., equipment and technologies) and intangible resources (e.g., knowledge, copyrights, and patents) that are normally used in the community.
- *Functional dimension*—refers to all the related operations, processes, methods, procedures, and functions that are used in the community.
- *Behavioral dimension*—refers to all those principles, governance rules, policies, and cultural issues that drive the behavior of the participants and community.

These endogenous elements and their related components and entities are listed in Table 3.

The exogenous elements focus on external interactions between the community and its surrounding environment. Exogenous interactions also include four dimensions, as follows:

- *Market dimension*—deals with interactions between the community and its clientele, competitors, and potential partners and sponsors. Part of this dimension handles related issues, such as the mission of the community, its value proposition, and joint identity.
- *Support dimension*—deals with the interaction between the community and those external support services (e.g., technical and financial services) that are mainly provided by third-party entities.
- *Societal dimension*—deals with general interactions (e.g., exchange, competition, and collaboration) and social actions (e.g., communication and volunteering) between the community and society (e.g., public and private organizations or other similar communities).
- *Constituency dimension*—deals with interactions between the community and its potential new members (e.g., attracting, recruiting, welcoming, directing, and encouraging).

All these external interactions and trans-organizational relationships might take place at three different levels:

○ *Community identity*—defining how the community presents itself to the surrounding environment and showing the position of the community in the environment in which it interacts with others.

○ *Interaction parties*—identifying the potential and interested entities (e.g., organizations, research centers, labs, and researchers) that the community can interact with.

○ *Interactions*—listing the various transaction types that can take place between the community and its interlocutors (e.g., partners, collaborators, and rivals).

The exogenous elements and their related components are shown in Table 4.

As can be seen in the Tables above, there are some similarities and overlaps among the elements in terms of structure and component parts because, in practice, these parts are not separable, and the related elements develop common tasks and practices.

**Table 3.** Endogenous elements of MCL.

| Endogenous Elements for MCL | | | |
|---|---|---|---|
| **Structural Dimension** | **Componential Dimension** | **Functional Dimension** | **Behavioral Dimension** |
| Participants <br><br> • They are volunteer <br> • They have different background <br> • They are autonomous <br> • They are distributed <br><br> Roles <br><br> • They are taken based on skills <br> • They are taken based on interests <br> • They are taken based on background <br><br> ➢ Managerial roles: <br>　• Identity controllers <br>　• Content controllers <br>　• Administrators <br>　• Supporters <br>　• Advisors <br><br> ➢ Technical roles: <br>　• Web designer and developers <br>　• Computer engineers <br>　• Technical operators <br>　• IT technicians <br><br> ➢ Participatory roles: <br>　• Experts <br>　• Ordinary members <br>　• Partners <br>　• Stakeholders <br><br> Roles Relationship <br><br> • Friendships <br> • Collaboration <br> • Communications <br> • Mentor-mentee <br> • Partnership <br> • Peer-to-peer <br> • Transactional <br> • Trusted <br><br> Community Typology <br><br> • Online collaborative learning <br> • Open, but may have access criteria <br><br> ➢ Type: <br>　• Strategic alliances <br><br> ➢ Size: <br>　• Unlimited | Resources <br><br> ➢ Technological Resources: <br>　• Websites <br>　• Platforms <br>　• Databases <br>　• CSCL tools <br>　• Internet <br>　• Hardware <br>　• Software <br><br> ➢ Human Resources: <br>　• Four types of groups: <br>　　- Ordinary groups <br>　　- Experts groups <br>　　- Advisors <br>　　- Managerial groups <br>　　- Technical groups <br>　• Three types of members: <br>　　- Top members <br>　　- Active members <br>　　- Inactive members <br><br> ➢ Knowledge Resources: <br>　• Knowledge <br>　• Information <br>　• Data <br>　• Repositories <br>　• Templates <br><br> ➢ Financial Resources: <br>　• Grants <br>　• Funds <br>　• Donations and aides <br><br> ➢ Community outcomes: <br>　• Developed knowledge <br>　• Developed skills <br>　• Developed competencies <br>　• Findings <br>　• Gained successes <br>　• Public awareness <br>　• Training services <br><br> ➢ Ontologies: <br>　• Community ontology <br>　• Participants' ontology | Processes <br><br> ➢ Fundamental processes: <br>　• Background processes <br>　　- Community establishment <br>　　- Community development <br>　　- Community dissolution <br>　• Management processes <br>　　- Membership management <br>　　- Profile management <br>　　- Task management <br>　　- Knowledge management <br>　　- Risk management <br>　• Execution processes <br>　　- Resource allocation <br>　　- Community evaluation <br>　　- Decision making <br>　• Supporting processes <br>　　- Documentation <br>　　- Configuration <br>　　- Verification <br>　　- Training <br><br> Procedures <br><br> ➢ Community building: <br>　• Goals establishment <br>　• Model selection <br>　• Resource provision <br>　• Rules setting <br>　• Foundation building <br>　• Facility provision <br>　• Member attracting <br>　• Contribution managing <br>　• Monitoring <br>　• Developing <br><br> ➢ Knowledge evolution approaches: <br>　• Knowledge creation is emphasized not knowledge acquisition <br>　• knowledge turns from tacit into explicit form <br>　• Knowledge quality assurance <br>　• Continual knowledge assessment <br>　• Learning from successful cases <br><br> ➢ Community operation handling: <br>　• Community uses common sense <br>　• Community uses a voting system <br>　• Experts' opinions are highly valued | Governance Model <br><br> • Self-governed community <br> • Non-hierarchical <br> • Decentralized <br> • Democratic <br> • Collaborative <br><br> Power within the Community <br><br> • Power is distributed <br> • There is no obligation <br> • Collaboration creates power <br><br> Rules and Policies <br><br> • Freely publish the findings <br> • Participants provide reliable materials <br> • Contents are written from the neutral viewpoint <br> • Contents are shared throughout the community <br> • Developed contents will be stored in a safe database <br> • Participants take full responsibility for their contributions <br> • Participants keep the community safe and respectful <br> • Participants receive equal opportunities <br><br> Culture <br><br> • Community orientation <br> • Outcome orientation <br> • Innovation <br> • Stability <br> • Creating value <br> • Following the rules <br> • Supporting others <br> • Criticizing ideas, not people <br> • Flagging bad behaviors <br><br> Constraint and Conditions <br><br> • Confidentiality constraints <br> • Internal normative constraints <br> • Ownership of the contents belong to the community |

**Table 4.** Exogenous elements for MCL.

| Exogenous Elements for MCL | | | |
|---|---|---|---|
| **Market Dimension** | **Support Dimension** | **Societal Dimension** | **Constituency Dimension** |
| Community Identity Level | | | |
| Community Mission<br>• External collaboration development<br>• lifelong learning encouragement<br><br>Community Profile<br>• Virtual learning community<br>• Connection building by online platforms (e.g., website, social media, ICT, email)<br><br>Market Strategy<br>• Boundary development<br>• Being served as an innovative library<br>• Being served as a community of practice<br>• Being served as an open knowledge lab | Community's Social Nature<br>• MCL is a not-for-profit community<br>• MCL can also provide monetary services<br>• MCL is an informal org<br>• MCL is a decentralized org<br>• MCL is a collaborative org<br>• MCL is a networked org<br>• MCL is an innovative org | Community legal Status<br>• MCL is a non-governmental community<br>• MCL is a self-governed community<br>• MCL is an informal community of learning<br>• MCL is an association of collaborators<br>• MCL cultivates deregulated learning<br>• MCL uses grants, charitable and philanthropic funds | Attracting and Recruiting Strategies<br>• Increasing community visibility (e.g., by social media)<br>• Using the word-of-mouth recommendations<br>• Developing the partnerships<br>• Keeping the community information up to date<br>• Taking easy approaches to inclusion and exclusion<br>• Using rewarding and ranking system (e.g., giving special access to the platform or information) |
| Interaction Parties Level | | | |
| Potential Customers/Clients<br>• Public/private organizations<br>• Educational centers<br>• Research centers<br>• Libraries<br>• Individuals<br>• Problem-solving markets<br>• Knowledge-intensive business services<br><br>Competitors<br>• Similar MCL projects (e.g., Wikipedia)<br><br>Potential Suppliers<br>• Universities<br>• Massive Open Online Courses (MOOC)<br>• Instructors | Financial Entities<br>• Public/private investors<br>• Sponsors/donators<br><br>Technical Entities<br>• IT companies/experts<br>• Network service provider<br>• Storage service provider<br><br>Informational Entities<br>• Universities<br>• Libraries<br>• Research institutes<br>• Experts/advisers<br>• Professional associations<br><br>Social Entities<br>• Living learning labs<br>• Research hubs | Governmental Organizations<br>• Educational and scientific org<br>• Intellectual property org<br>• Advisory Councils<br><br>Private Sectors<br>• Knowledge-intensive business services<br>• Laboratories<br><br>NGOs<br>• Education charities<br>• Advocacy NGOs<br><br>Interested Entities<br>• Businesses<br>• Learning and training services<br>• Consulting services | Organizations<br>• Public/private org<br>• Public/private Institutes<br>• Public/private business<br>• Enterprises<br>• Corporations<br>• Libraries<br>• Laboratories<br>• Research centers<br>• Consult services<br><br>Individuals<br>• Experts<br>• Professionals<br>• Inexpert |
| Interaction level | | | |
| Customer/Client Interactions<br>• Engagement<br>• Collaboration<br>• Consultation<br><br>Competitor Interactions<br>• Knowledge exchanging<br>• Partnering<br>• Joining<br>• Supporting<br><br>Supplier Interactions<br>• Joining<br>• Partnering<br>• Supporting | Support/Service Acquisition<br>• Financial support<br>• Technological support<br>• Information service<br>• Consulting service<br>• Training service<br>• Researching service<br>• Donation service<br>• Coaching actions<br>• Alliances<br><br>Agreement Establishment<br>• Contracting<br>• Dealing<br>• Community affiliation | Political Relations<br>• Making/developing relationships between community and org<br><br>Social Relations<br>• Developing the collaboration<br>• Sharing the findings<br><br>Knowledge sharing<br>• Face-to-face/directly<br>• Social media<br>• Broadcasting<br><br>Seeking Support<br>• Consultation | Member Searching<br>• Advertising<br>• Talking to friends and co-workers<br>• Sending invitation/solicitation<br>• Participants can bring in new faces<br>• Conduct workshops<br><br>Joining Process<br>➢ Sending application by applicant<br>➢ Evaluating the application by the community, and:<br>  • Accepts the application, or<br>  • Rejects the application, or<br>  • Requests correction |

## 4.3. Assessment Approaches

Once the objectives are defined, structural and behavioral models are adapted, and specific internal and external components are identified and customized, the MCL community can then proceed to the operation phase. In this phase, the community needs to assess its organizational changes. For that reason, the assessment phase helps the community to monitor its success over time, drive instruction and learning, control learning progress, and boost motivation [8]. The assessment can take place at two levels: namely, at an individual level and a community level:

- *Assessment at the individual level*—an approach to teaching and learning that creates feedback about the learners' performance, progress, and/or achievements. It also indicates whether or not the learners have achieved learning outcomes or met the goals of their individualized programs [36].

- *Assessment at the community level*—a systematic process for analyzing the strengths and weaknesses of the community in relation to its performance, and the factors that affect performance. Additionally, assessment at this level can obtain valid information about the community's processes, work environment, and organizational structure [37].

In this work, apart from the assessment processes, methods, tools, and strategies, attention is only given to those assessment indicators that can potentially be used at both levels. The proposed indicators are listed in Table 5.

**Table 5.** Proposed assessment indicators.

| Assessment Indicators for MCL Communities | |
|---|---|
| **Indicators for Individual Performance Assessment** | **Indicators for Community Performance Assessment** |
| Input Indicators<br>Having good socio-economic conditions<br><br>- Having equipped environment<br>- Enough training courses<br>- Matches of mentors and mentees<br>- Enough related contents<br>- Useful contents<br>- Reliable contents<br>- Learner's prior knowledge<br>- Learner's aptitude<br>- Learner's readiness<br>- Trainer's quality and impacts<br>- Consultation hours<br><br>Activity Indicators<br><br>- Enough motivation factors<br>- Active contribution<br>- Active collaboration<br>- Effective collaboration<br>- Enough time for doing tasks<br>- Minimized barriers to learning<br><br>Outcome Indicators<br><br>- Improved knowledge<br>- Increased learning skills<br>- Improved collaboration skills<br>- Developed relationships<br><br>Impact Indicators<br><br>- Increased civic awareness<br>- Promoted collaborative learning<br>- Being able to solve related problems | • Organizational<br>  - Having enough capacity to deliver services<br>  - Having effective coordination techniques<br><br>• Environmental<br>  - Having a safe and healthy atmosphere<br>  - Enough resources for collaboration<br><br>• Structural<br>  - Enough application of learning standards<br>  - Clarity in the division of responsibilities<br><br>• Behavioral<br>  - Adequate user commitment and retention<br>  - Enough coaching<br><br>• Admission<br>  - Active recruitment/employment rate<br>  - Enough influential stakeholders<br><br>• Social<br>  - Effective communication<br>  - Active contribution<br><br>• Functional<br>  - Enough network productivity<br>  - Matches of training courses to future career plans/needs<br>  - Internal promotions Vs. external hire<br><br>• Technological<br>  - Equipped environment<br>  - Enough technical administrators<br><br>• Economical<br>  - Increase in revenues/profits<br>  - Enough financial supports |

As can be seen in Table 5, in order to better understand the type and specifications of the proposed indicators for individual performance assessment, they are placed into four main categories:

- *Input indicators*—focusing on resources that are needed for creating the learning program;
- *Activity indicators*—focusing on activities and operations that are used in the learning program;
- *Output indicators*—focusing on the expected outcome/results that can be achieved by the learning program in the long and the short run; and
- *Impact indicators*—focusing on learning program contribution to higher-level strategic plans.

At the community level, performance assessment indicators are grouped under 9 considered dimensions of collaboration: namely, organizational, environmental, structural, behavioral, admission, social, functional, technological, and economical. It should be noted that each MCL community needs to define its specific performance indicators, based on its objectives and conditions. A well-developed set of performance indicators (at both levels) can make appropriate links among strategy, operation, and ultimate value creation.

Another important issue in the context of community assessment is dealing with big data. MCL (due to the size of the community and the number of learners) faces the challenge of handling the huge amount of content generated, uploaded, and shared [17,38]. On top of that, there is abundant knowledge, information, and data with different degrees of quality (e.g., non-relevant or of low quality), each coming from a different location [8]. To effectively cope with this problem, the literature proposes different potential techniques (e.g., natural language processing [38] and deep learning [39]) or technologies (e.g., smart systems and software) [40], but it remains a huge challenge.

In order to diminish the distribution of unhealthy content (e.g., disinformation and fake news) and promote the quality of shared materials in MCL communities, this work proposes a mixed-method model (MAM-MCL) [8] that can help the learners involved to measure the quality and reliability of shared content through a multi-user and multilevel evaluation approach. As illustrated in Figure 2, MAM-MCL comprises four main steps. These steps are briefly explained in the following:

- *Step 0—sharing a piece of content*: in this step, the community members share a piece of content with others in the community.
- *Step 1—assessment by technology*: in this step, the shared content will be preliminarily checked and then filtered by means of technology (e.g., an AI software filtering system). The checked content: (a) might be rejected when it is useless or is against community rules, (b) might be referred back to the learner when it needs some changes or modification, or (c) might be passed on to the next step, when it can proceed for further assessment.
- *Step 2—assessment by moderator(s):* in this step, the received content (e.g., controversial cases and suspicious content) from step 1 is checked by the moderator(s) for quality assurance (manually, by computer, or both). Similar to the previous step, the received content: (a) might be rejected, (b) might be referred back to the learner, or (c) might proceed for further assessment. In this case, the checked content will then be categorized according to pre-defined fields, classes, and topics. Afterward, the content is sent to the respective assessment level (Step 3). In cases where the checked content "is not controversial", it should then be sent to the individual level. If the checked content "is controversial" (e.g., having ethical, cultural, or critical issues), it should preferably be sent to the community level. This is because it is assumed that the community—in comparison with individuals—is better placed to assess and make decisions about such controversial content.
- *Step 3a—referral to the individual level*: when the content is recognized as "not controversial", it is sent by a moderator to the individual level for quality assessment.
    - \> Assessment by ordinary participants at the individual level: they assess the content that is recognized as neither "controversial", nor "scientific and professional". Here, the assessed content could be rejected, accepted, or referred to the community level (ordinary participants) for further assessment.

> Assessment by expert participants at the individual level: they assess the content that is found "not controversial", but "scientific and professional". The assessed content at this stage could be rejected, accepted, or referred to the community level (expert participants) for further assessment.

- *Step 3b—referral to the community level*: when the considered content is realized as "controversial", it should be referred to the community level.

  > Assessment by ordinary participants at the community level: they assess the content that is categorized as "controversial" but "not scientific and professional".

  > Assessment by expert participants at the community level: they assess the content that is grouped as "controversial" and "scientific and professional".

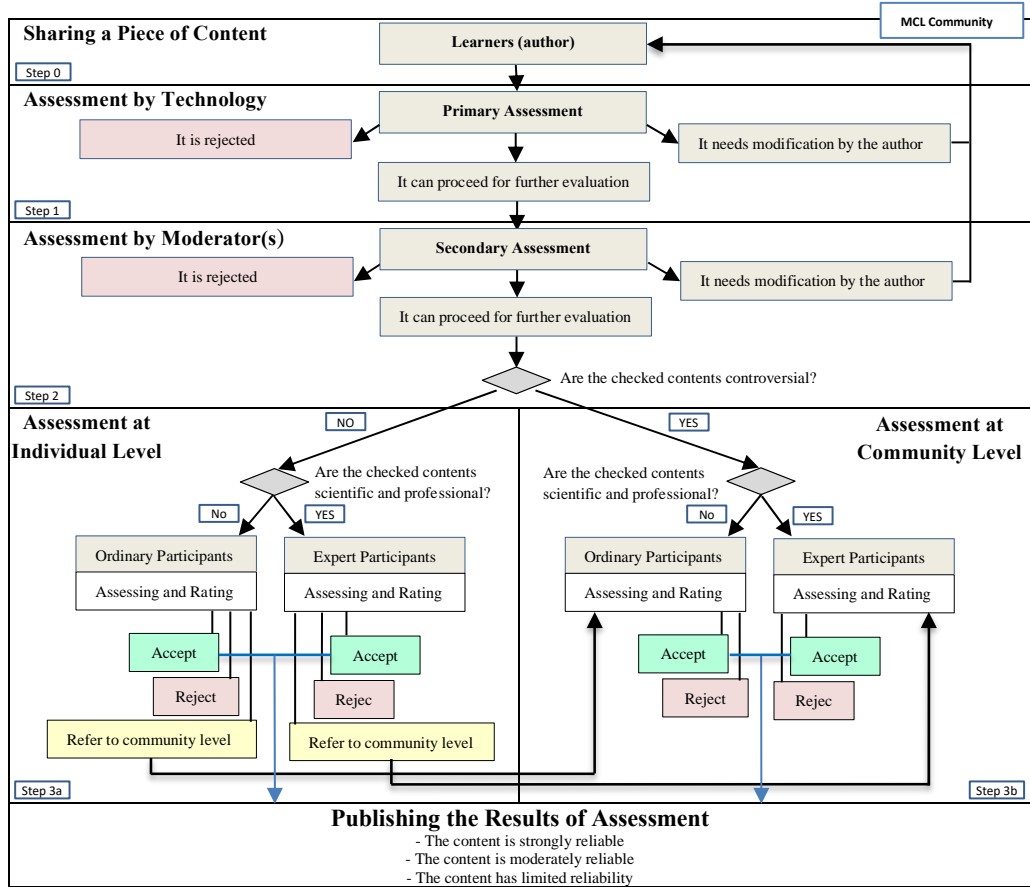

**Figure 2.** Proposed MAM-MCL for assessing the reliability of shared content in the MCL community.

When the results of assessments at both levels show that the content is accepted, it should then be published for use by the community. It is believed that by including all types of community members in assessment at different stages, the results will not only represent a common collaboration but can also provide a certain degree of content quality assurance.

## 5. Case Studies

To demonstrate and evaluate the effectiveness and successfulness of the meta-governance framework in supporting the establishment, operation, and development of MCL communities, we proceeded with the instantiation of the framework in two case studies and one MCL illustration.

*5.1. Case Study 1*

The meta-governance framework is used in the ED-EN HUB [41], an Erasmus + project co-financed by the European Union and developed by a consortium of 8 institutions from 5 different European countries. The ED-EN HUB project aims to improve the quality of education (focusing but not limiting itself to vocational education and training) through the consolidation and systematization of the education enterprise. This international cooperation alliance aims to allow the development of tools and methodologies toward the creation of synergies between educational institutions and enterprises.

ED-EN HUB uses a collaborative and knowledge-oriented platform that provides a supportive environment for the training and learning hub, where a large number of scattered but enthusiastic trainers and learners from different backgrounds come together to promote their knowledge and competencies. The participants in the hub attempt to adopt new ways and develop more scenarios for sharing their knowledge, experiences, and ideas, which leads to a higher level of understanding, qualification, and performance.

For the application of the meta-governance framework to the ED-EN HUB project and its platform, the framework was first customized and then evaluated. Therefore, the partners and stakeholders of the project, through three rounds of (developed) questionnaires, tried to evaluate the adequacy, feasibility, and effectiveness of the framework to be implemented on the platform. Via the first round of questionnaires, the partners selected a number of components (from part 1 of the framework) that were suitable from their viewpoint to be considered for the platform. It was considered to be a general analysis of the proposed dimension of collaboration and their project's respective features and components. Via the first step of the second round of questionnaires, the technical partners evaluated the feasibility of the selected components and features (from the first round of questionnaires), based on the available budget and capabilities that they had. Then, in the second step of feasibility evaluation (phase), by taking into account the selected components and features compared with (the defined) system functions, the technical partners developed the second specified questionnaire. The selected components and features from this phase of evaluation were lastly evaluated by those partners and stakeholders (via the third round of questionnaires) to ensure that the components and features were effective and efficient enough to be implemented on the platform. On the one hand, all these three phases of evaluation helped the partners to appropriately adapt the framework to the platform. On the other hand, the evaluations assisted them in better developing the platform's functions. Additionally, the results gained from the three-phase evaluation process show that the meta-governance framework is fairly adequate, feasible, and effective when used in the ED-EN HUB and its platform, even though some of the addressed components and features in the framework still need to be modified and customized, based on the objectives and requirements of the project.

*5.2. Case Study 2*

The meta-governance framework is also being used for the ENHANCE project, which is co-funded by the Erasmus + program of the European Union [42]. This project focuses on developing a knowledge-transfer framework and mechanisms, as well as contributing to strengthening the skills and training expertise of Tunisian and Moroccan universities in three targeted topics: namely, maintenance engineering, production engineering, and quality engineering (MPQ) for inciting and assisting the transition of both partner countries to the era of Industry 4.0. There are several considered objectives for the project to be fulfilled, for example, to create a lifelong eLearning (LeL) platform for practitioners, and to develop a learning framework addressing MPQ 4.0 skills, etc.

Concerning each objective, several functions are needed, as follows:

- *Objective A*—creating a lifelong eLearning (LeL) platform for practitioners, one that is able to support collaboration and learning within large networks of participants:
  - > Function 1—the platform (which is usually video-based in some way) provides an online collaborative environment, where universities, teachers, and students

can share their content. This function can be supported by several parts of the meta-governance framework, such as structural and behavioral models, endogenous elements, and the content assessment method.

> Function 2—teachers can construct classes, deliver courses, upload videos, and assign and grade quizzes and homework assignments. This function can receive support from some parts of the meta-governance framework, such as learning and performance assessment.

> Function 3—students can collaborate on finding courses and consuming course content. This function can be directed by some parts of the meta-governance framework, such as structural and behavioral models and endogenous elements.

> Function 4—the platform provides a built-in method for students to practice their new skills and receive feedback from teachers. This function can be guided by some parts of the meta-governance framework, such as the behavioral model.

> Function 5—the classes on the platforms are self-paced, either in part or in full. This function can be supported by some parts of the meta-governance framework, such as the structural model.

- *Objective B*—developing a learning and knowledge transfer framework, addressing MPQ skills for Industry 4.0:

> Function 1 (in maintenance engineering)—data acquisition regarding equipment, technologies, and functions. This function can be supported by some parts of the meta-governance framework, such as exogenous elements and the content assessment method.

> Function 2 (in production engineering)—a decision support system for the evaluation of continuous production plans. This function can be derived from certain parts of the meta-governance framework, such as performance assessment.

> Function 3 (in production engineering)—data analytics for business intelligence and value creation out of production data. This function can be steered by some parts of the meta-governance framework, such as the content assessment method.

> Function 4 (in quality engineering)—real-time or near-real-time quality control in manufacturing. This function can be supported by certain parts of the meta-governance framework, such as performance assessment.

In the ENHANCE project, a similar approach and process were used to evaluate the meta-governance framework for adequacy, feasibility, and effectiveness assurance. The results of evaluations and several critical discussions prove that the meta-governance framework (after making the necessary modifications and development according to the project goals and circumstances) has high potential to be applied to the project as a supporting and directing guideline.

As another example of the application of the meta-governance framework to this case study, it should be mentioned that the project is creating and developing several digital innovation hubs (DIHs) in Tunisian and Moroccan universities. The DIH model provides an ecosystem that stimulates the adoption of advanced digital technologies to facilitate collaboration among the involved stakeholders [43]. The meta-governance framework is used (in some ways) as a guide to support the establishment, operation, and development of the DIHs.

*5.3. Mass Collaborative Learning Illustration*

The meta-governance framework was also used in an illustration of the MCL community. This illustration was developed in a master's thesis [44] that presents a community of cooks (both professional and non-professional) who come together to learn new things about cooking. In this community, by providing different self-created video clips from their own dishes and sharing them within the community, the cooks try to (a) create a comprehensive list (video collection) of their national and international dishes, (b) share

their related knowledge and experiences with their peers, (c) increase the level of information about the quality and quantity of ingredients, and (d) raise the level of knowledge about healthy diets, balanced diets, and nutrition. For this purpose, a digital platform was developed to support the community members in acquiring, creating, improving, and sharing diverse culinary knowledge.

The meta-governance framework was used in this MCL illustration to support the creation and development of a collaboration platform for cooks. The platform provides a social virtual environment that, by reaping the advantages of the most advanced technologies (e.g., online discussion forums and voting systems), helps users to make connections, develop collaboration and transactions, share their content (e.g., video clips, recipes, comments, and tips) with others, and evaluate and vote on the contributions of others. For example, the framework provides the GloFood community with some guidance and solutions regarding the assessment of shared materials.

In this case, the meta-governance framework was successfully used as guidance for the presentation and characterization of the cooking community. As such, the framework steered and helped to define the main concepts, structures (organizational and behavioral), components (internal and external), functions (collaboration, interactions), roles (e.g., organizer, executive, and content controller), and participants (managerial, expert, ordinary) used in the illustration. As a result, a potential, promising, and supportive environment was developed for those interested cooks who were eager to learn new things collaboratively. The successful application of meta-governance to the illustration case study is considered as evidence of framework validation.

## 6. Limitations

Despite the tremendous progress, notable achievements, and positive results that MCL has obtained over the years for learners, communities, and societies, realizing this concept of a body of knowledge still faces several challenges. These relate to:

- The novelty of the MCL concept;
- The complexity of its underlying processes;
- The interdisciplinary nature of the method;
- The fact that the organizational structure and associated mechanism of MCL is still evolving;
- The still insufficient evidence regarding the successful application of MCL in various fields;
- The challenges of dealing with the diffusion of fake or low-quality information/knowledge;
- The fact that the process of MC in the learning community is not yet clearly formalized; and
- The fact that there are some ambiguities regarding the process of stimulating people to join the community and keep them motivated to contribute.

Furthermore, this approach of collective learning is viewed and presented disparately, from different perspectives, by researchers in each discipline. As a result, this promising complementary approach to learning and accessing an amazing breadth of experience has not yet been revealed to all people around the world.

## 7. Conclusions

MCL, as a complementary method of learning that stands alongside the traditional (currently used) educational system, tries to provide a set of informal learning opportunities for both the general public who cannot attend the actual classrooms (for any reason) and for those people willing to explore extra learning practices. The main principles of MCL communities (openness, social production, participatory learning, collaboration, peer-reviewing, and acting globally) can be adjusted based on the nature of the problems being tackled. MCL is a complex system, one that emerges in different forms in the diverse application domains and contains many facets, the proper understanding of which needs contributions from varied disciplines. After reviewing the literature and receiving inspiration from a collection of relevant frameworks, the authors of this work propose

a meta-governance framework, addressing a set of models, components, and methods that can potentially support and direct the process of implementation, operation, and development of concrete cases of MCL, developed elsewhere.

The meta-governance framework proposed in this study is a general model that needs to be adapted before it can be applied to specific MCL cases. The application of the meta-governance framework in three case studies showed its potential when designing new learning communities. Due to the novelty of the MCL concept, as well as the inadequacy of the available reports about its successful application in different domains, this study faced some limitations in finding the requisite resources.

By application of the proposed meta-governance framework in three case studies (two EU research projects and one illustrative community of cooks), the primary practical experimentation provided us with a relatively acceptable indication of the potentiality and applicability of the framework for use in other learning communities. This conclusion is drawn as the people involved in these three case studies showed their willingness to apply the framework to their projects, indicating their acknowledgment and assurance of the competence of the framework. Furthermore, the successful application of the framework in the case of the community of cooks could support the creation and development of a collaborative learning platform and its governance principles. Similarly, the primary promising results gained from the framework evaluation (adequacy, feasibility, and effectiveness) and its application in the case of the two EU projects show that there is a good chance that the meta-governance framework can be applied, developed, and validated in other related scenarios.

In future work, we will try to apply the MGF-MCL to other case studies, with the aim of appropriately developing and more thoroughly validating the framework.

**Author Contributions:** Conceptualization, M.Z. and L.M.C.-M.; methodology, M.Z. and L.M.C.-M.; validation, M.Z., L.M.C.-M. and J.S.; formal analysis, M.Z. and L.M.C.-M.; investigation, M.Z. and L.M.C.-M.; resources, M.Z., L.M.C.-M. and J.S.; data curation, M.Z. and L.M.C.-M.; writing—original draft preparation, M.Z. and L.M.C.-M.; writing—review and editing, M.Z. and L.M.C.-M.; visualization, M.Z., L.M.C.-M. and J.S.; supervision, L.M.C.-M.; project administration, M.Z., L.M.C.-M. and J.S.; funding acquisition, L.M.C.-M. and J.S. All authors have read and agreed to the published version of the manuscript.

**Funding:** Fundação para a Ciência e Tecnologia (project UIDB/00066/2020) and European Commission ERASMUS + under grant n°. 619130-EPP-1-2020-1-FR-EPPKA2-CBHE-JP ENHANCE and grant n° 2020-1-FR01-KA202-080231 ED-EN HUB.

**Institutional Review Board Statement:** Not applicable.

**Informed Consent Statement:** Not applicable.

**Data Availability Statement:** The data has been presented in main text.

**Acknowledgments:** This work was supported by the Center of Technology and Systems (CTS-UNINOVA).

**Conflicts of Interest:** The authors declare no conflict of interest.

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
