# Peer review of "Meta-Governance Framework to Guide the Establishment of Mass Collaborative Learning Communities"

_computers, doi:10.3390/computers11010012_

Round 1
Reviewer 1 Report
The analyzed article is interesting for the scientific community. The study covers a relevant topic to contribute to the advancement of the state of the question. The article is well structured. However, it is necessary to adapt the research to the current context derived from Covid-19. Therefore, I advise the authors to read the suggested documents. With the purpose of incorporating literature on the current pandemic situation and other innovative ways to carry out training actions in these times of academic and health uncertainty. Furthermore, it is important to add after the conclusions the main practical implications derived from this study.
Corell-Almuzara, A.; López-Belmonte, J.; Marín-Marín, J.-A.; Moreno-Guerrero, A.-J. COVID-19 in the Field of Education: State of the Art. Sustainability 2021, 13, 5452. https://doi.org/10.3390/su13105452
Moreno-Guerrero, A., Soler-Costa, R., Marín-Marín, J., & López-Belmonte, J. (2021). Flipped learning and good teaching practices in secondary education. [Flipped learning y buenas prácticas docentes en educación secundaria]. Comunicar, 68, 107-117. https://doi.org/10.3916/C68-2021-09
Author Response
Dear Reviewer
Thank you very much for reviewing our manuscript and providing us with very helpful comments and directing remarks. In the following, you can find our reaction to your comments. (Please find the attachment as well)
It is necessary to adapt the research to the current context derived from Covid-19
Despite the advantages of traditional face-to-face and in-person learning, it faces some limitations or difficulties, for example, travel time and cost, fees, scheduling, low flexibility, etc. In contrast, as in MCL the collaboration and interactions predominantly take place over the Internet, it not only hinders involvement with such limitations and difficulties but also MCL can be used as a flexible learning solution when face-to-face learning is impossible or at risk. Particularly, in the current condition of the Covid-19 pandemic which is causing alterations in educational systems and (temporary and compulsory) shifting away from the classroom to alternative modes of learning (e.g., virtual) [18], MCL can respond to the current pandemic and create a long-term positive impact by providing some support for every student/citizen around the world. As an illustration, MCL can provide an innovative and suitable environment with specific features (e.g., safe, customizable, affordable, conventional, flexible, and accessible) equipped with modern technologies (e.g., video conferencing, blogs, discussion forums, or platforms) that through which a large number of interested people (e.g., teachers, students, and even general public) at regional, national, or international level can engage in collaborative learning practices (e.g., sharing learning materials, discussion) throughout the crisis and even then after. [L 136 - 152]
It is important to add after the conclusions the main practical implications derived from this study
By application of the Meta-Governance framework in three case studies (one master thesis and two EU projects), the primary practical experimentation provided us a relatively acceptable indication of the potentiality and applicability of the framework to be used in other learning communities. This conclusion is made as these three case studies showed their willingness to apply the framework to their projects, indicating their enough assurance of framework competence. Furthermore, the successful application of the framework to the master thesis could support the creation and development of the collaborative learning platform used. On the other side, the primary promising results gained from framework evaluation (adequacy, feasibility, and effectiveness) and application by the two EU projects show that there are so many chances for the Meta-Governance framework to be applied, developed, and validated by other related cases. [L 815 - 826]
Added Reference to the Manuscript
[18] Corell-Almuzara, A.; López-Belmonte, J.; Marín-Marín, J.-A., Moreno-Guerrero, A.-J.: COVID-19 in the Field of Education: State of the Art. Sustainability 2021, 13(10), 5452. https://doi.org/10.3390/su13105452

Reviewer 2 Report
Mass Collaborative Learning Communities (MCL) is an exciting topic with multiple implications in distance online learning. However, the current manuscript has several weaknesses that need to be addressed:
L737: A quick look at the references table reveals at least 16 self-citations of authors. This is can be partially explained by the fact that the manuscript presents a work compiled in a PhD thesis. However, among them there are several self-citations of works dating more than 10 years ago. In any case, I believe this number is too high, both in absolute and relative terms.
L58-60: Not all MOOCs are examples of mass collaboration, in fact one can argue that the opposite is the case: most MOOCs are based on individual tasks, plagued from user isolation, where social interactions are minimal, hence completion rates are very low.
Reversely, beyond MOOCs, WoW and Scratch, the manuscript should engage more with contemporary literature and adjacent fields, educational theories and practices focusing on social online environments that support mass collaborative learning with socio-constructivist and connectivist underpinnings [1].
One very relevant and well-established community-centered theoretical model that should be taken into account is the community of inquiry framework, a staple for deep meaningful learning in online settings [2].
As authors mention mixed and virtual reality, they should also acknowledge the field of virtual worlds, social virtual reality or multiuser virtual environments [3,4] where the concept of online community has been prevalent and the issues of governance and moderation of large online communities has also been discussed [5–7].
Additional adjacent fields for consideration and contrast are communities in open source software [8], social media [9] and citizen science [10].
L81: Please explain what is meta-governance and why it is relevant or important in this context. Is effective community governance not sufficient?
L540: inspect the Figure’s spelling for typos, e.g. connet
L583: A Discussion section normally succeeds Results presentation. Case studies seem to be better suitable for the Results section.
L603-4: Although the framework aims at the mass scale, how is an implementation at a much smaller scale helpful? Or does it apply universally to all learning communities of any size? How do findings prove the effectiveness of the model?
L604-622: The case study’s process is clear however, concrete results and outputs of the process are missing to support the derived benefits.
L624: It is unclear how the framework is connected to community’s Functions. It appears as a post-hoc analysis of a community with already decided components and procedures.
L677: Can the illustration as a post-hoc analysis of an established community be considered as evidence of a model? Again basic, concrete data about the implementation are missing, e.g. how many cooks participated in this community?
I suggest adding a section with the study’s limitations.
References
- Anderson, T.; Dron, J. Three generations of distance education pedagogy. Int. Rev. Res. Open Distrib. Learn. 2011, 12, 80, doi:10.19173/irrodl.v12i3.890.
- Garrison, D.R.; Anderson, T.; Archer, W. The first decade of the community of inquiry framework: A retrospective. Internet High. Educ. 2010, 13, 5–9, doi:10.1016/j.iheduc.2009.10.003.
- Mystakidis, S.; Berki, E.; Valtanen, J.-P. Deep and Meaningful E-Learning with Social Virtual Reality Environments in Higher Education: A Systematic Literature Review. Appl. Sci. 2021, 11, 2412, doi:10.3390/app11052412.
- Pellas, N.; Kazanidis, I.; Konstantinou, N.; Georgiou, G. Exploring the educational potential of three-dimensional multi-user virtual worlds for STEM education: A mixed-method systematic literature review. Educ. Inf. Technol. 2017, 22, 2235–2279, doi:10.1007/s10639-016-9537-2.
- Mystakidis, S. Motivation Enhancement Methods for Community Building in Extended Reality. In Augmented and Mixed Reality for Communities; Fisher, J.A., Ed.; CRC Press: Boca Raton, 2021; pp. 265–282.
- O’Connor, E.A. Developing Community and Building Knowledge Online Using a Virtual Reality Environment and Student-Created Videos. J. Educ. Technol. Syst. 2017, 46, 343–362, doi:10.1177/0047239517736874.
- Warren, I.; Palmer, D.; King, T.; Segrave, S. Second Life and the role of educators as regulators. In Proceedings of the ASCILITE 2008: Hello! Where are you in the landscape of educational technology?; ASCILITE: Melbourne, 2008; pp. 1079–1089.
- O’Mahony, S.; Ferraro, F. The Emergence of Governance in an Open Source Community. Acad. Manag. J. 2007, 50, 1079–1106, doi:10.5465/amj.2007.27169153.
- Li, K.C.; Wong, B.T. The Opportunities and Challenges of Social Media in Higher Education: A Literature Review. SN Comput. Sci. 2021, 2, 455, doi:10.1007/s42979-021-00857-5.
- Hinojosa, L.; Riedy, R.; Polman, J.; Swanson, R.; Nuessle, T.; Garneau, N. Expanding Public Participation in Science Practices Beyond Data Collection. Citiz. Sci. Theory Pract. 2021, 6, 11, doi:10.5334/cstp.292.
Author Response
Dear Reviewer
Thank you very much for reviewing our manuscript and providing us with very helpful comments and directing remarks. In the following, you can find our reaction to your comments. (please find the attachment as well)
L737: A quick look at the references table reveals at least 16 self-citations of authors. This is can be partially explained by the fact that the manuscript presents a work compiled in a PhD thesis. However, among them there are several self-citations of works dating more than 10 years ago. In any case, I believe this number is too high, both in absolute and relative terms.
The fact is that three references dating more than 10 years ago are associated with the master thesis of the first author. As they are related in some ways to the topic of the Ph.D. thesis (of the first author), they are then used in this study to make a link and provide some support and evidence for the content. However, based on your suggestion two of them (24 and 36) are now deleted.
L58-60: Not all MOOCs are examples of mass collaboration, in fact one can argue that the opposite is the case: most MOOCs are based on individual tasks, plagued from user isolation, where social interactions are minimal, hence completion rates are very low.
One very relevant and well-established community-centered theoretical model that should be taken into account is the community of inquiry framework, a staple for deep meaningful learning in online settings [2].
The success that mass collaboration has gained over the last years in the learning domain is in fact noticeable. The literature shows that mass collaboration has morphed and expanded the physical, virtual, and intellectual boundaries of the learning environments [10]. As such, mass collaboration has had successful applications (at different levels and to different degrees) in diverse learning contexts, for example:
- Massive Open Online Courses (MOOCs), which are free online courses where the learning contents are delivered to any person who wants to take a course. In this model of learning that is designed for large numbers of geographically dispersed students, they can practice learning individually (in personal tasks, thus not really mass collaboration) and through community interactions (in interactive courses, featuring a form of mass collaboration).
- World of Warcraft (WoW), which is a massively multiplayer online role-playing game that facilitates learning through gamification. WoW creates a community of players where they can play with others in temporary groups. In this collaborative space, learning occurs when the learner needs and wants it. Therefore, in the context of problem-solving, there are opportunities to receive the answers (from other players or more experienced peers) to a question or obtain advice quickly.
- Scratch, which is a free programming language and online community where scratchers/learners can create their own interactive stories, games, and animations. Scratch promotes problem-solving skills, self-expression, collaboration, and creative teaching and learning. There is a discussion page with multiple forums mainly used for chatting, helping (with coding), creating, sharing, and learning together [11].
- SAP Community Network (SCN), which is a community of software users, developers, consultants, mentors, and students who use the network to get help, share ideas, learn, innovate, and collaborate [12].
- Community of Inquiry (Col) framework is a community of learners and instructors who share a virtual space, technology-reliant environment, rule-based interaction, and course-dependent learning objectives that resulted from the interaction of the perceptions of social, cognitive, and teaching presences. [13].
- ePals global community, which is an example of social online learning that provides the needed tools (platform) and meeting places to build a worldwide community of learners, global citizens who can share ideas, practice communication, and offer help and guidance [14].
[L 62 - 94].
As authors mention mixed and virtual reality, they should also acknowledge the field of virtual worlds, social virtual reality or multiuser virtual environments [3,4] where the concept of online community has been prevalent and the issues of governance and moderation of large online communities has also been discussed [5–7].
Additional adjacent fields for consideration and contrast are communities in open source software [8], social media [9] and citizen science [10].
MCL as a kind of open-source community and flexible learning environment can potentially take place based on diverse environments such as mixed reality, virtual reality [2], social virtual reality environments (immersive virtual worlds or multi-user virtual environments) [3], multi-user 3D interactive environments, and 3D multi-user virtual worlds (e.g., social virtual worlds, open-source virtual worlds, collaborative virtual learning worlds) [4] [L 39 - 44]
L81: Please explain what is meta-governance and why it is relevant or important in this context. Is effective community governance not sufficient?
In this work, the Meta-Governance framework is considered as a comprehensive, organized, and specified structure for being used by MCL initiatives. The proposed Meta-Governance framework is a generic and conceptual structure intended to highlight the main concepts, factors, and connections associated with MCL communities. The Me-ta-Governance framework integrates ideas from different governance styles (e.g., participatory or democratic governance, corporate governance, and network governance) and then consolidates the key components, attempting to introduce a structure (with mixed dimensions) that can serve as a guide for supporting the MCL communities. For example, there should be a set of processes, functions, and activities (e.g., assessments, social participation, interactions) in each MCL community that should be properly carried out to meet and accomplish the goals of the community. Hereupon, the Me-ta-Governance framework is developed, aiming to portrait, guide, and streamline such particular works. [L 346 - 357]
L540: inspect the Figure’s spelling for typos, e.g. connet
The Figure is modified
L583: A Discussion section normally succeeds Results presentation. Case studies seem to be better suitable for the Results section.
The Section title is changed
L603-4: Although the framework aims at the mass scale, how is an implementation at a much smaller scale helpful? Or does it apply universally to all learning communities of any size? How do findings prove the effectiveness of the model?
The Meta-Governance framework and containing elements and features are defined to be used at a mass scale, however, it is possible to be used for learning communities of any size. The fact is that since Meta-governance is a dynamic framework, so it should be then adapted according to the objectives, requirements, and conditions of the concrete case. [L 358–362]
L604-622: The case study’s process is clear however, concrete results and outputs of the process are missing to support the derived benefits.
The results gained from the three-phase evaluation process show that the Meta-Governance framework is fairly adequate, feasible, and effective to be used in the ED-EN HUB and its platform even though some of the addressed components and features in the framework need to be modified and customized based on the objectives and requirements of the project. [L 677 - 681]
In the ENHANCE project, a similar approach and process were used to evaluate the Meta-governance framework for adequacy, feasibility, and effectiveness assurance. The results of evaluations and several critical discussions prove that the Meta-Governance framework (after making the needed modification and development according to the project goals and circumstances) has high potential to be applied to the project as a supporting and directing guideline. [L 728 - 733]
L624: It is unclear how the framework is connected to community’s Functions. It appears as a post-hoc analysis of a community with already decided components and procedures.
For example, there should be a set of processes, functions, and activities (e.g., assessments, social participation, interactions) in each MCL community that should be properly carried out to meet and accomplish the goals of the community. Hereupon, the Meta-Governance framework is developed, aiming to portrait, guide, and streamline such particular works. [L 353 -357]
L677: Can the illustration as a post-hoc analysis of an established community be considered as evidence of a model? Again basic, concrete data about the implementation are missing, e.g. how many cooks participated in this community?
The Meta-Governance was used in the MCL illustration to support the creation and development of a collaboration platform for cooks. The platform provides a social virtual environment that by reaping the advantages of the most advanced technologies (e.g., online discussion forums and voting systems) helps the users to make connections, develop collaboration and transactions, share their contents (e.g., video clips, recipes, comments, and tips) with others, and evaluate and vote others' contributions. For example, the framework provides the GloFood community with some guidance and solutions regarding the assessment of shared materials.
In this case, the Meta-Governance framework was successfully used as guidance for the presentation and characterization of the cooking community. As such the framework steered and helped to define the main concepts, structures (organizational and behavioral), components (internal and external), functions (collaboration, interactions), roles (e.g., organizer, executive, and content controller), and participants (managerial, expert, ordinary) used in the illustration. As a result, a potential, promising, and supportive environment is developed for the interested cooks who are eager to learn new things collaboratively. The successful application of Meta-governance to the illustration is considered as evidence of framework validation. [L 754 -770]
I suggest adding a section with the study’s limitations
Despite the tremendous progress, notable achievements, and positive results that MCL has obtained over the years for learners, communities, and societies, this body of knowledge is still faced with several challenges. For example:
- The novelty of the MCL concept,
- The complexity of its underlying processes,
- The interdisciplinary nature of the method,
- The organizational structure and associated mechanism of MCL are still evolving,
- There is insufficient evidence about the successful application of MCL in various fields,
- The process of MC in the learning community is not clearly formalized, and
- There are some ambiguities about the process of stimulating people to join the community and keep them motivated to contribute.
Furthermore, this approach of collective learning is viewed and presented from different perspectives by the researchers of each discipline. As a result, this promising complementary approach of learning and amazing experience has not yet been revealed to all people around the world. [L 768 -785]
Added References to the Manuscript
[2] Mystakidis, S.: Motivation Enhancement Methods for Community Building in Extended Reality. In Augmented and Mixed Reality for Communities; Fisher, J.A., Ed.; CRC Press: Boca Raton, 2021; pp. 265–282
[3] Mystakidis, S.; Berki, E.; Valtanen, J.-P. Deep and Meaningful E-Learning with Social Virtual Reality Environments in Higher Education: A Systematic Literature Review. Appl. Sci. 2021, 11, 2412, doi:10.3390/app11052412.
[4] Pellas, N.; Kazanidis, I.; Konstantinou, N.; Georgiou, G.: Exploring the educational potential of three-dimensional multi-user virtual worlds for STEM education: A mixed-method systematic literature review. Educ. Inf. Technol. 2017, 22, pp. 2235–2279, doi:10.1007/s10639-016-9537-2.
[13] Garrison, D.R.; Anderson, T.; Archer, W.: The first decade of the community of inquiry framework: A retrospective. The Internet and Higher Education, 13(1–2), pp. 5-9, 2010. doi:10.1016/j.iheduc.2009.10.003
[14] Bernadette Dwyer B.: Teaching and Learning in the Global Village: Connect, Create, Collaborate, and Communicate. The Reading Teacher, 70(1), pp. 131–136, 2016. doi:10.1002/trtr.1500

Round 2
Reviewer 2 Report
Authors have addressed all identified issues in a satisfactory manner by expanding and improving significantly the quality of the manuscript rendering it fit for publication.
Author Response
Thank you very much for your comments.
Please find the attached (revised version).
Best regards
Majid Zamiri
